# Considerations for Intravenous Anesthesia Dose in Obese Children: Understanding PKPD

**DOI:** 10.3390/jcm12041642

**Published:** 2023-02-18

**Authors:** James Denzil Morse, Luis Ignacio Cortinez, Brian Joseph Anderson

**Affiliations:** 1Department of Anaesthesiology, University of Auckland, Park Road, Auckland 1023, New Zealand; 2División Anestesiología, Escuela de Medicina, Pontificia Universidad Católica de Chile, Santiago 8331150, Chile; 3Department of Anaesthesia, Auckland Children’s Hospital, Park Road, Private Bag 92024, Auckland 1023, New Zealand

**Keywords:** pharmacokinetics, pharmacodynamics, pediatrics, drug dosing, allometry, obesity, anesthesia

## Abstract

The intravenous induction or loading dose in children is commonly prescribed per kilogram. That dose recognizes the linear relationship between volume of distribution and total body weight. Total body weight comprises both fat and fat-free mass. Fat mass influences the volume of distribution and the use of total body weight fails to recognize the impact of fat mass on pharmacokinetics in children. Size metrics alternative to total body mass (e.g., fat-free and normal fat mass, ideal body weight and lean body weight) have been proposed to scale pharmacokinetic parameters (clearance, volume of distribution) for size. Clearance is the key parameter used to calculate infusion rates or maintenance dosing at steady state. Dosing schedules recognize the curvilinear relationship, described using allometric theory, between clearance and size. Fat mass also has an indirect influence on clearance through both metabolic and renal function that is independent of its effects due to increased body mass. Fat-free mass, lean body mass and ideal body mass are not drug specific and fail to recognize the variable impact of fat mass contributing to body composition in children, both lean and obese. Normal fat mass, used in conjunction with allometry, may prove a useful size metric but computation by clinicians for the individual child is not facile. Dosing is further complicated by the need for multicompartment models to describe intravenous drug pharmacokinetics and the concentration effect relationship, both beneficial and adverse, is often poorly understood. Obesity is also associated with other morbidity that may also influence pharmacokinetics. Dose is best determined using pharmacokinetic–pharmacodynamic (PKPD) models that account for these varied factors. These models, along with covariates (age, weight, body composition), can be incorporated into programmable target-controlled infusion pumps. The use of target-controlled infusion pumps, assuming practitioners have a sound understanding of the PKPD within programs, provide the best available guide to intravenous dose in obese children.

## 1. Introduction

Total body weight dosing in obese children contributes to dose errors because the contribution from the fat mass portion of the body composition is not acknowledged. Although it is recognized that fat mass may influence pharmacokinetic parameters such as volume of distribution (V) or clearance (CL) [1], that the effect of fat mass is drug-specific [1], that weight-based dosing is a contributor to dose inaccuracies and that obesity influences disease processes, there are few practical dose recommendations for obese children [2,3]. A smorgasbord of body weight scalers (e.g., total body weight, body surface area, ideal body weight, lean body mass, adjusted body weight, body mass index, fat-free mass, allometry) have been used to determine dose in the obese individual [4]. There is often confusion as to which metric is best suited for an individual child and that metric may change between phases of anesthesia (e.g., lean body mass for propofol induction dose and total body weight for maintenance dose rate) [5,6,7]. Consequently, recommendations for any size scaler are tempered by expert opinion that presumes dose in the obese child will be determined by better pharmacokinetic understanding [3,8,9].

Total intravenous anesthesia (TIVA) in infants and children has spurred investigation of pharmacokinetic and pharmacodynamic models to improve dose estimation [10]. At the heart of pediatric pharmacokinetic models are two covariates, size and age: these account for major components of parameter (e.g., clearance, volume) variability [11,12]. Size can be standardized to a 70 kg person using allometric theory [13]; age can be used as a measure to quantify clearance maturation [14]. These two covariates commonly feature in target-controlled infusion pumps, allowing the anesthetic practitioner to manually enter this information into the pump program [15,16]. Body composition, particularly fat mass [17], is another important covariate in the obese child, but has been poorly investigated [18]. 

We review the concepts behind dose determination for pediatric anesthesia infusion and attempt to rationalize the quantification and impact of fat mass on dose determination.

## 2. Pharmacokinetic Concepts to Determine Dose

The pharmacokinetic parameter volume of distribution (V) is used to determine the loading dose that achieves a desired target concentration (TC) for a simple one-compartment model (Equation (1)), while clearance (CL) determines the maintenance dose or infusion rate (Equation (2)).
(1)Loading Dose=V×TC
(2)Maintenance Dose Rate=CL×TC

Most body weight scalers demonstrate a nonlinear relationship between clearance and size, a relationship that is evident in both obese and lean individuals [1]. The rate of clearance increase slows as size increases; consequently dose, when expressed as per kilogram of total body weight, is invariably excessive. Total body weight is a poor size scaler and ideal body weight (IBW), which has a non-linear relationship to clearance (i.e., rate of clearance increase slows as size increases), is currently the only alternative body weight scaler to total body weight (TBW) [19] mentioned in the British National Formulary for Children [20]. However, there are five published methods available for calculation [21] and poor understanding exists of when and how to calculate IBW among clinicians [22]. Further, IBW is not the best scaler for all drugs [9]. Clinicians would be better served to understand the pharmacokinetic–pharmacodynamic principles that determine dose, rather than choosing an arbitrary size scaler (e.g., IBW) that is, albeit better than total body weight that is commonly used to determine a dose that is too big.

Infusion dose for children undergoing intravenous anesthesia is based on the target concentration strategy [23]. The target concentration is that which achieves a target effect. Pharmacokinetic parameters are used to determine the dose that achieves a target concentration (Figure 1) [24]. Covariates such as size, age and organ function influence pharmacokinetic parameters (CL, V). Fat mass also has an impact on these pharmacokinetic parameters, but fat mass is rarely included in dose calculation for either obese or lean children. 

## 3. Dosing Concepts in the Child

The principles behind dose estimation involve an understanding of the pharmacokinetic parameters of clearance and volume. Although these two parameters contribute the most observed dose variability in children [12], fat mass has an influence on both these parameters, even in non-obese individuals. Pharmacokinetic principles for dose in a typical child require explanation. 

### 3.1. The Association between Weight and Dose

Drugs are commonly dosed per kilogram of total body weight. This is because the pharmacokinetic parameters (e.g., CL, V) that determine dose are based on size. Once the impact of size is understood, then other covariate influences (e.g., age related changes, organ dysfunction or obesity-associated physiology changes) can be evaluated. Drug dose calculations are commonly made as per kilogram; this assumes a linear relationship between dose and TBW (Equation (3)).
(3)Dose=DoseSTD×(TBWWTSTD)
where a standard dose (Dose_STD_) is considered appropriate for a person of a standard weight (WT_STD_ e.g., 70 kg). However, maintenance doses expressed as mg/kg, as in Equation (3), are commonly observed to be too small in children when compared to adults. For example, propofol infusion rates to maintain a target concentration of 3 mg/L are higher in children than adults [25]. These observations query linear assumptions about dose and weight and point to why the linear approach is not a suitable general method for drug dosing in children [6].

Clearance is the key parameter for the determination of maintenance dose. Prediction using adult human clearance values using the linear per kilogram model results in an underprediction of more than 10% at body weights less than 47 kg. This underprediction increases as size decreases and approaches 50% in infants 1–2 years of age (i.e., around 10 kg) [26]. Clearance is commonly reduced in neonates due to lack of maturation of elimination processes (characterized using age), so dose predictions in babies may be appropriate, but this is serendipitous rather than supportive of the linear per kilogram model.

### 3.2. Use of PKPD to Determine Dose

The goal of pharmacological treatment is achievement at a specific target effect. A pharmacodynamic (PD) model (e.g., E_MAX_ or Hill equation [27], Figure 1) is used to predict the target concentration known to be associated with a desired target effect. Population pharmacokinetic (PK) and pharmacodynamic (PD) parameter estimates, as well as covariate information, are used to predict time concentration and concentration effect values in a specific patient.

#### 3.2.1. The Target Concentration

The target concentration strategy is used widely to determine drug dose [28,29] in anesthesia. This approach is used almost instinctively by pediatric anesthesiologists who use target-controlled infusion systems. These devices target a specific plasma or effect site concentration in a typical child, and that concentration is assumed to have a typical target effect (Figure 1). Covariate influences such as patient age or weight are manually entered into the TCI pump program. Adverse effects are monitored (e.g., bradycardia and hypotension). The ideal target concentration achieves therapeutic effect (e.g., anesthesia depth) without untoward adverse effects. 

An effect site target concentration has been estimated for many drugs used in anesthesia, analgesia and sedation. The relationship between propofol concentration and effect (bispectral index, BIS) has been used to identify a target concentration of 3 mg/L (3 µg/mL), and this realizes a BIS therapeutic range of 40–60 in children [30,31]. Concentration–response relationships for remifentanil [32,33], clonidine [34], sevoflurane [35], acetaminophen [36], dexmedetomidine [37,38], alphaxalone [39] and albuterol (salbutamol) [40,41] have been established, enabling target effect and consequent concentration estimation. 

#### 3.2.2. Dose Calculation Using Compartment Models

Dose calculation invariably uses the volume of distribution (V) and clearance (CL) along with the required target concentration (TC) to achieve the desired effect (Equations (1) and (2)) [42]. A single compartment is often inadequate for characterizing the time concentration profile, and further compartments are required to describe drug disposition satisfactorily. A drug is usually administered into a central compartment (V1) and then redistributes to peripheral compartments (e.g., V2, V3; Figure 1). Calculations used for a one-compartment model may not be appropriate for many anesthetic drugs that are characterized using multi-compartment models. The loading dose may be too small if based on V1 for a drug described using multiple compartments where redistribution is happening during loading dose administration (e.g., dexmedetomidine) [43]. Pharmacokinetic compartment models describing drugs administered intravenously often have different estimations of the central compartment volume [44]. The central volume of distribution estimated using lean children results in clinicians instinctively overshooting the targeted plasma concentration (Cp), and this may be unsuitable with obese children using TBW as a size scaler because fat mass is unaccounted for.

Drug infusion rate at a steady state is determined by clearance, but many drugs used in anesthesia practice distribute to peripheral compartments and steady state may not be achieved during infusion duration. Dose adjustment is required to achieve a consistent, stable concentration associated with its desired effect. Propofol PK, for example, are usually described using a three-compartment mammillary model. Manual dose regimens require consecutive rate step downs after defined time intervals to accommodate redistribution among compartments. These step downs are illustrated by the ‘10-8-6’ rule for propofol infusion (loading dose of 1 mg/kg followed immediately by an infusion of 10 mg/kg/h for 10 min, 8 mg/kg/h for the next 10 min and 6 mg/kg/h thereafter) in adults [45]. Similar propofol manual infusion regimens have been described for children [25,30]. Target-controlled infusion (TCI) pumps make infusion adjustments at 10 s intervals rather than 10 min intervals, enabling fine tuning.

## 4. Allometry

While total body weight may not be the best size scaler, when pharmacokinetic parameters are estimated using total body weight with allometry, then it appears to be a reasonable body scaler for many drugs (e.g., propofol [46]). Use of allometry accounts for the nonlinear relationship between size and clearance. Allometry describes the relation between the size of an organism or system and aspects of its morphology, physiology and life history. The relationship between physiological traits (e.g., metabolic processes) and structural components (e.g., blood volume) and size has been used to scale pharmacokinetic parameters. Allometric theory is used to explain the nonlinear relationship between clearance and size [13,47]. 

The log of basal metabolic rate plotted against the log of body weight produces a straight line with a slope of ¾ across species, with size changes that are 18 orders of magnitude. Fractal geometry is used to mathematically explain this phenomenon known as allometry [48,49]. Total drug clearance may be expected to scale to weight with an exponent of ¾ (Equation (4)) [50], so that clearance in a child can be predicted from that in an adult of standard weight 70 kg (WT_STD_):(4)CLchild=CLadult×FAGE×(TBWWTSTD)3/4

Clearance maturation occurs in the first year of life and a function describing this maturation is required (Figure 2, F_AGE_) during that period. Propofol clearance in obese adults [51] and non-obese adults and in children [15,52,53,54,55] and in obese children [56] is best described using allometry with TBW as the size descriptor rather than FFM, LBM or IBM. Allometry computation is relatively easy for practitioners and can be managed on an application of a cellular phone. The only variable required is weight and knowledge of a standardized clearance in an adult; height is not a required variable. 

Clearance is less than might be expected from total body weight with the linear per kilogram model and this deviation increases with total body weight. The other body weight scalers shown in Figure 2 (body surface area, lean body mass, fat-free mass) are also curvilinear in nature. 

## 5. Size Scalers for the Obese Child

It is understood that 75% of excess weight in obese children is fat mass, and the remainder is lean mass [18]. It is thought that increases in fat mass primarily alter distribution of lipophilic drugs and increases in lean mass alter drug clearance, but there is a lack of evidence supporting these assumptions for most drugs [18]. Instead of good quality evidence, investigators have used an assortment of size scalers to empirically explain the contribution of fat mass for individual drugs [7].

### 5.1. Size Scaling and Obesity in Anaesthesia

Body fat has minimal metabolic activity, however, fat mass may have an indirect influence on both metabolic and renal functions that is independent of its effects due to increased body mass. Obesity is associated with increased morbidity, and this effect that is independent of body size can influence clearance. This morbidity, exemplified as liver dysfunction for example, may influence clearance, volume and protein binding. These effects of morbidity may appear as differences in body composition even when there is no clear marker of morbidity. Anesthesia practitioners have used a number of size scalers to estimate dose in obese individuals. 

### 5.2. Body Mass Index

Body mass index (BMI) is the commonest marker used to define obesity, but it is not commonly used to predict dose in children because it fails to distinguish between adipose tissue and lean muscle mass. Further, BMI in children must be interpreted with reference to age and sex, while percentiles around the median are required to define grades of obesity. 

### 5.3. Lean Body Mass

It has been claimed that lean body mass (LBM) (often used interchangeably with lean body weight (LBW) and fat-free mass (FFM)) is the optimal size scaler for most drugs used in anesthesia, including opioids and anesthetic induction agents [7,60,61,62]. This argument has also been put forward for drugs used outside of anesthesia with the explicit assumption that clearance is linearly related to LBM. The use of lean body mass without allometric scaling appears to be a good predictor of dose for remifentanil [63]. It has also been proposed to be better than TBW for both propofol infusion and loading dose calculation in adults [64]. The advantages and merits of using LBM or FFM have been reviewed with the surprising conclusion that LBM is a good predictor of drug dose for all drugs [65]. This extension to all drugs remains unproven [9] and back-to-back comparison with other size scalers has been rarely undertaken. Despite a lack of validation, the LBM size scaler [57] has been incorporated into some propofol infusion target controlled pumps. The use of LBM calculated using the James formula [57] results in biologically implausible values in obese adults of short stature (Figure 2). Methods used prevent excess dose in pumps used for adults include calculation of a fictitious height [66] or incorporating limits on the maximum weight allowed by the TCI program [67]. 

### 5.4. Fat-Free Mass

Fat-free mass (FFM) is an alternative but similar size metric to LBM. FFM comprises muscle, bone, vital organs and extracellular fluid, but does not include lipids in CNS, bone marrow and cell membranes [68,69]. These additional lipids comprise only a small part of TBM (3% to 5%) and as a result FFM is often used interchangeably with LBM in clinical practice. One clinical advantage of FFM over LBM is that it avoids failure in obese individuals of short stature [70,71,72]. 

Fat-free mass (FFM) can be predicted from sex, height and total body weight (Equation (5)).
(5)FFM=WHSMAX×HT2×[TBW(WHS50×HT2+TBW)]
where WHS_MAX_ is the maximum FFM for any given height (HT, m) and WHS_50_ is the TBW value when FFM is half of WHS_MAX_. For men, WHS_MAX_ is 42.92 kg·m^−2^ and WHS_50_ is 30.93 kg·m^−2^ and for women WHS_MAX_ is 37.99 kg·m^−2^ and WHS_50_ is 35.98 kg·m^−2^ [58].

The extrapolation of the FFM formula developed in adults (Equation (5)) has been described for children as young as 3 years [73], but does not include neonates and infants. Fat mass increases over the first 9 months of postnatal life [74] and the impact of body composition on pharmacokinetics in infants remains unquantified. Girls have an FFM similar to that predicted from adults based on height, weight and sex. Boys have lower than predicted FFM until around the onset of puberty when muscle mass increases and FFM approaches that described in adults. FFM has been proposed as the scaler in a model for remifentanil that is applicable to both adults and children [16].

### 5.5. Ideal Body Weight

Ideal body weight (IBW) has been recommended as the preferred metric for maintenance dosing of a number of drugs, e.g., benzodiazepines (diazepam [75], midazolam [76]), morphine [77] and neuromuscular blocking drugs, such as vecuronium [78], rocuronium [79,80] and cisatracurium [81]. The relationship between IBW and clearance is certainly curvilinear, but it is not the best metric for all drugs. The search for the “best” body size predictor has revealed that more mass than expected from IBW may be required for some drugs. The addition of 40% of the weight above IBW (the “excess weight”) for propofol infusion calculation has been suggested [82]. Rocuronium dosing improved when 20% of the excess weight was added to IBW [79]. The use of IBW for all drugs does not ring true. Clearance of propofol was best predicted using TBW and allometry [15,30,46,53]. 

## 6. A Universal Size Scaler 

A simplified formula exists to determine LBM based on the observation that a mean of 29% of the excess weight carried by an obese child is lean tissue [7,83,84]. This can be expressed using Equation (6):(6)LBM=IBM+0.29×(TBM−IBM)

A better size scaler would be drug-specific. The size metric common for all drugs requires recognition of fat-free mass (FFM) and the added contribution from fat mass. Such a flexible size scaler could be incorporated into dose calculations for use in both manual and target-controlled infusion devices. 

### 6.1. Normal Fat Mass

Any size scaler must account for fat mass and must be applicable to children of all weights. There seems little value in using total body weight for children who are lean and then switching to an alternative size scaler in those children classified as obese. Adding a fraction of fat mass to FFM has been used to estimate the mass that best describes size based on allometric scaling theory [42]. This mass has been called normal fat mass (NFM) [85]. Fat mass can be calculated from TBW and FFM (Equation (7)): FAT = TBW − FFM(7)

Normal fat mass (NFM) is then FFM plus a little bit more. That little bit more is defined by the parameter Ffat. The fraction of FAT (Ffat) that contributes to the structural (V) or functional (CL) size is specific to each drug (Equation (8)): (8)NFM=FFM+Ffat×FAT

If Ffat is estimated to be zero then FFM is the predicted size ideal for dosing calculation (e.g., remifentanil), while if Ffat is 1 then size is better predicted by TBW (e.g., propofol). The use of normal fat mass (NFM) based on allometric theory and partition of body mass into fat and fat-free components provides a principle-based approach applicable to predicting size and body composition effects on pharmacokinetics of all drugs in children and adults. NFM requires the determination of Ffat that can be applied to volume or clearance and this has only been determined for a handful of drugs (Table 1). 

A negative value for Ffat suggests organ dysfunction associated with fat, and this pathology is reported in morbidly obese adults (e.g., fatty liver disease). Dexmedetomidine was noted to have a negative value for Ffat applied to CL in morbidly obese adults [89]. Although Ffat possibly increases with lipid solubility, this has not yet been demonstrated. The precision of fat mass estimates when calculated using population modelling estimates has been questioned because of limited population numbers studied [99].

### 6.2. Limitations of NFM for IV Dosing

A major limitation of NFM use is that its calculation is impractical in the busy clinical theater environment; calculation is more complex than FFM or ideal body weight that pediatricians find onerous [100]. These size scalers will never be used if not programmed into target-controlled infusion (TCI) pumps. Use without a TCI pump would require a reference table such as that for dexmedetomidine in Table 2.

Table 2 is a simplification and only shows dexmedetomidine (Ffat = 0) dose in four individuals of similar age but different total body weights. There is no obvious dose trend that relates dose to increases in weight. Although the per kilogram dose decreases with increasing weight, we cannot eyeball these doses and claim a 10% dose reduction for each 10 kg weight increase, for example. Tables such as this would be impractical because they concern only one drug with one age of fixed height. The larger assortments of weights, heights and ages are not shown, and a large number of charts would be required for each drug in common usage. Further, dose is titrated down as drug redistributes through compartments during infusion [45]. Steady state where clearance determines infusion rate is not achieved even after 1 h in this example using dexmedetomidine. 

The practicing anesthesiologist requires either an accessible hand-held computing device to make calculations or infusion pumps that are programmed with NFM parameters to make dose calculation automatic. 

## 7. Application of NFM Principles for TCI to the Obese and Non-Obese Child

Once the impact of fat mass on pharmacokinetic parameters (CL, V) has been evaluated, then those pharmacokinetic parameters can be used in all children, lean or obese. It is not necessary to change to a different size scaler simply because the patient fulfills criteria that determine obesity.

### 7.1. Maintenance/Infusion Dose

The difference in drug clearance between an adult and a child is predictable from NFM used in conjunction with allometry (Equation (9)):(9)CLCHILD=CLADULT×(NFMCHILDNFMADULT)34

Maintenance dose rates (oral doses or continuous infusions) can be calculated based on the target concentration (Equation (10)) [101]:(10)Infusion Dose Rate=Clearance×Target Steady State Concentration

The target concentration in children and adults is often similar [102] and this allows the relationship between doses in children and adults can be predicted (Equation (11)):(11)DoseRateCHILD=DoseRateADULT×(NFMCHILDNFMADULT)34

### 7.2. The Dose–Clearance Mismatch Explained

The use of allometry in programmed infusion pumps for anesthetic drugs will correct for clearance changes with size [103]. Use of NFM with allometry, rather than TBW with allometry, will result in better programmable pharmacokinetic parameter sets for obese children. Maintenance dose rates (per kilogram) will be lower in obese children than in lean children.

This dose reduction can be demonstrated using a simulation for propofol with the NFM size descriptor and allometry. Propofol is a drug where Ffat is one (i.e., TBW and NFM are the same), revealing the pharmacokinetic principles without the additional impact of fat mass. When propofol is delivered at a set rate (10 mg/kg/h), concentrations are higher and bispectral index (BIS) is lower in the obese child (6 years 50 kg, BMI 37.81 kg/m^2^) compared to the lean child (6 years 20 kg, BMI 15.12 kg/m^2^). The steady state concentration, dictated by clearance, is achieved at the same time but there are differences between concentrations and consequent effect (Figure 3). This is because clearance, expressed as per kilogram, is lower in the obese child. Dose, expressed as per kilogram, also requires a similar reduction. Dose reduction, per kilogram, will be similar to the nonlinear changes in clearance (Figure 2). Use of NFM without allometry would result in a dose that is too large because the curvilinear nature of clearance changes with size are unaccounted for. The use of NFM with allometry, irrespective of the value for Ffat, is a better option, since knowing the Ffat of each different drug allows adaptation of the scaler according to the drug’s physical properties.

There is no dose–clearance mismatch. Dose is dictated by clearance. Clearance can be expressed as a nonlinear function of size using allometry. The size metric is NFM rather than TBW.

### 7.3. Loading Dose

An infusion duration of four half-lives is required to reach 93.74% of steady state plasma concentration (Cp) (Figure 3). Consequently, a loading dose is often used to rapidly achieve an effect site concentration (Ce) associated with anesthesia. Volume of distribution is the key parameter used to determine loading dose.

The volume of distribution of a drug may depend on its physicochemical properties. There are drugs whose apparent volume of distribution may be independent of fat mass, e.g., digoxin has similar volumes in obese and non-obese subjects [104]. Some drugs show no change in disposition with weight loss in obese subjects [105], while others are extensively associated with weight loss (e.g., benzodiazepines such as diazepam [75,106,107]). Associations between volume of distribution and lipid solubility are poor [85]. Diazepam has been heralded as an example of a lipid soluble drug with a large volume of distribution. Although the larger volume of distribution of diazepam in obese adults has been attributed to its fat solubility [75], a lower diazepam binding inhibitor in obese patients [108] could also account for this larger volume of distribution because a reduced diazepam binding inhibitor leads to higher tissue binding. 

NFM can be used as the size descriptor for loading dose, but the estimation of Ffat for volume will differ from that estimated for clearance.

#### 7.3.1. Loading Dose Using a One-Compartment Model

Pharmacokinetic parameter estimates for most drugs used in anesthesia are well investigated. Use of reported volume of distribution to estimate loading dose is tempered by several additional considerations. It is the concentration in the effect compartment that drives response and there is a delay between plasma and effect compartments. In order to achieve rapid achievement of effect compartment concentration, high concentrations in plasma are required to ‘drive’ drug into the effect compartment. Those concentrations in plasma are often associated with adverse effects. In addition, estimates of V1 are invariably estimated in well children. The impact of fat mass on these estimates is not commonly assessed. 

#### 7.3.2. Loading Dose for an Obese Child

The size scalers predicted to best describe volume of distribution are many and differ for every drug. A consistent scaler would be normal fat mass. The volume of distribution can be corrected for normal fat mass where Ffat is a unique parameter for each drug. For example, the Ffat estimate for remifentanil is 0 [16], while that for propofol is 1 [15,46]; dexmedetomidine has an Ffat of 0.293 [46]. The use of NFM, if Ffat is known, makes estimation of the loading dose for the obese child easy because the volume of distribution is known. 

However, most drugs used for infusion in anesthesia conform to multicompartment models. The central volume parameter V1 may be unsuitable to calculate loading dose due to both uncertainty around its estimation and redistribution to other compartments. Dose estimation based on V1 may be too low. An alternative volume, the volume of distribution at steady state (Vss), may result in a dose that is too high.

The loading dose is used to target a concentration at the effect site, not plasma, and there is a time delay between peak plasma concentration and peak concentration at the effect site. Multi-compartment models require a volume of distribution that accounts for dynamic changes in compartments during dosing. 

#### 7.3.3. The Volume of Distribution at Maximum Effect

The other pharmacokinetic parameters used to describe disposition (V2, V3, Q2, Q3) in multicompartment models also influence dose determination. The delay between plasma and effect site is also part of the dynamic changes that occur during administration of the loading dose. The time to peak effect (T_PEAK_) is dependent on clearance and the effect site equilibration half-time for a one compartment model, but intercompartment clearances also have influence on T_PEAK_ in multicompartment models. At a submaximal dose, T_PEAK_ is independent of dose because maximum effect is not reached. At supramaximal doses, maximal effect will occur earlier than T_PEAK_ and persist for a longer duration because of the shape of the pharmacodynamic (PD) sigmoidal concentration–response relationship; the achieved concentration is on the upper flat part of the curve (Emax). The T_PEAK_ concept has been used to calculate optimal initial bolus dose [109]. The volume of distribution (Vpe) at the time of peak effect site concentration (C_PEAK_) is calculated and used (Equation (12)):(12)$$\mathrm{Vpe}=\frac{\mathrm{Dose}}{\mathrm{Concentration}\left(T_{\mathrm{PEAK}}\right)}$$

Loading dose can then be calculated as (Equation (13)):(13)$$\mathrm{Loading}\;\mathrm{Dose}=C_{\mathrm{PEAK}}\times \mathrm{Vpe}$$

The simulations in Figure 4 demonstrate determination of the volume of distribution at the time of peak effect site concentration (Vpe), i.e., at peak effect. This example considers a 6-year-old 50 kg child given dexmedetomidine 1 µg/kg; clearance and volume parameters were expressed using NFM with allometry; clearance assumed FFM (Ffat = 0), and Ffat of 0.293 for V [46]. Clearance (expressed as per kilogram) is lower in the obese child, T_PEAK_ delayed compared to the child not obese, plasma concentrations higher with lower BIS scores.

#### 7.3.4. Consideration of Adverse Effects

Loading dose estimation can be determined using pharmacokinetic knowledge with NFM as a scaler. Use of the Vpe parameter allows calculation of a dose that achieves a target concentration and consequent target effect. However, anesthesia drug dose commonly results in adverse effects such as hypotension and bradycardia.

Adverse effects occur with rapid infusion of dexmedetomidine. A bolus dose of dexmedetomidine 0.49 µg/kg IV given over 5 s in children (5–9 years) caused a maximum median heart rate decrease of 20 beats per minute and a maximum median mean arterial pressure (MAP) increase of 12.5 mmHg, which happened 100 s after the bolus dose [110]. Dexmedetomidine is commonly administered more slowly over 15–20 min in children compared to adults (10 min) to maintain cardiovascular stability and avoid rapid change in blood pressure or pulse.

These relationships between concentration, effect (BIS) and cardiovascular effects can be illustrated using PKPD simulation to show the physiological effects of rapid and slow dexmedetomidine loading dose administration (Figure 5). Parameter estimates for the PK model were from Morse and colleagues [87]. Pharmacodynamic (PD) models for blood pressure were from Potts and colleagues [111] and heart rate changes are from Perez-Guille and colleagues [112]. An adult concentration–sedation model using bispectral index [38], a measure of anesthesia depth, was used as illustrative of sedation score. The dexmedetomidine equilibration half-time (T_1/2_keo) has been estimated as 3–6 min for sedation effect [37,113] and 9.9 min for centrally mediated vasodilation [111]. Rapid administration of dexmedetomidine 0.9 µg/kg in a 6-year-old (50 kg, BMI 37.81 kg/m^2^) child achieved acceptable sedation but at the cost of rapid pulse rate falls and blood pressure rises. When infusion time is over 20 min, a larger dose (1.1 µg/kg) is required for the same level of sedation (BIS 73), but cardiovascular changes are less fluctuant (Figure 5). 

## 8. Practical Considerations

There is a curvilinear relationship between clearance and weight. Infusion doses should reflect that relationship. Ideal body weight has been proposed as an appropriate size scaler for use in obese children. However, IBW calculation is not easy, and although it may describe a curvilinear relationship with clearance, it is neither drug-specific nor does it distinguish between clearance or volume.

The use of allometric scaling is a better option. Although calculation of allometric total body mass (TBM) or FFM (Equations (4) and (5)) is not something that is performed by most practitioners using mental arithmetic, tables such as Table 3 could be made accessible in the operating room. These computed masses (TBM representing Ffat 1 and FFM representing Ffat 0), corrected using allometric ¾ exponent scaling, could be substituted for actual total body weight in a manual infusion pump to predict clearance in the individual child. Both size scalers (TBM and FFM) have larger calculated masses than TBW in children to compensate for increased clearance (when expressed as per kilogram) and a lesser mass in those larger individuals, consistent with the curvilinear nature of clearance changes with weight (Figure 2). There remain difficulties with the use of such tables. Body composition in children differs from adults and drugs may not conform to use of TBM (e.g., propofol) or FFM (e.g., remifentanil); most drugs require a size standard somewhere in between, and the Ffat value (Equation (8)) remains unknown. 

Fat mass has influence on both clearance and volume and fat mass is present in children, even those who are lean. Body composition changes with age, as does the fat mass [74]. The definition of obesity in children is based on body mass index (BMI) changes with age (Growth Charts—Clinical Growth Charts (cdc.gov)). The BMI index is less in childhood than infancy. There is a nadir at 6–7 years and it increases with subsequent age. While adult obesity might be classified a grade 1 (BMI > 30 kg/m^2^), grade 2 (BMI > 35 kg/m^2^) and grade 3 (BMI > 40 kg/m^2^), child obesity is graded depending on the percentile above the median for that particular age group; overweight (>85th percentile), obese (>95th percentile) and severely obese (120% of 95th percentile or >35 kg/m^2^). A crude but practical method to determine a size scaler that uses readily measured variables (total body weight and height) might be a calculated weight for the pump that comprises:(A)Overweight: use total body weight less 5%(B)Obese: use total body weight less 10%(C)Severely obese: use total body weight less 20%

This simplification enables the use of total body weight only and maintains the curvilinear relationship between weight and clearance. This scaling is superior to current linear per kilogram calculations and appears reasonable for anesthesia infusion, but it is only acceptable because current predicted concentrations and effect measures have large between-subject variability [29,30]. Cerebral monitoring using processed EEG is commonly used to guide dose because of this known PKPD variability.

## 9. Conclusions

It is clear that the dose (mg/kg) in children with obesity is less than that presumed using linear scaling. Loading dose remains difficult to predict. PKPD modelling can be used to predict dose in obese children. PKPD models can be used to consider adverse effects after rapid infusion and suggest changes to infusion rate. However, there are other factors not considered: cardiac output, disease states, patient anxiety. Drug interactions (e.g., propofol and midazolam) also influence each other drug’s pharmacokinetic and pharmacodynamic response [114,115]. The anecdote of the classic examination question asked of anesthesia examination candidates ‘What dose of thiopentone?’ has a real sting in its scorpion-like tale.

Expert opinion continues to presume that the dose in the obese child will be determined by pharmacokinetic understanding [3,8]. While this opinion is certainly valid, there are a number of thorns on this rose of an opinion. Anesthetic drug pharmacokinetics and pharmacodynamics of many drugs used in children have been clarified and the use of NFM as a size scaler has merit, but the computations required to calculate dose in obese, or even non-obese, children are beyond most clinicians. The implementation of target-controlled infusion pumps that are programmable with these models (equations) enables better dosing in the obese. Models for propofol [15], remifentanil [16], dexmedetomidine [87], acetaminophen [116] and ibuprofen [116] incorporate both allometry and NFM and can be programmed into TCI pumps. Until NFM is defined for other drugs, the use of TBW with allometry captures the decreasing per kilogram dose with increasing size.

## Figures and Tables

**Figure 1 jcm-12-01642-f001:**
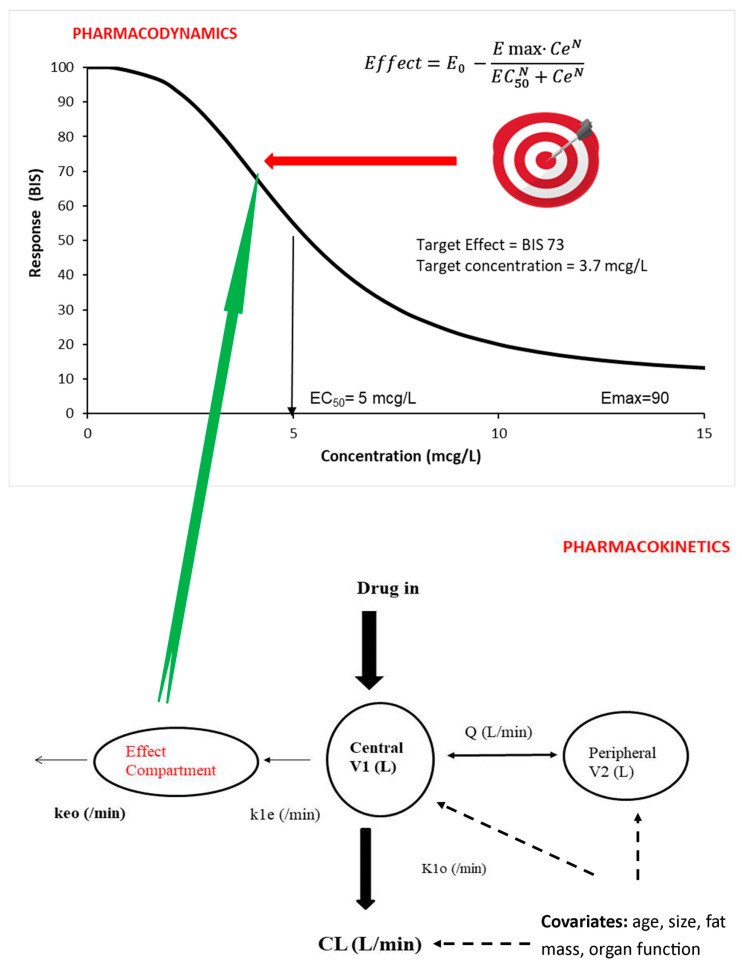
Principles behind the target concentration strategy are shown diagrammatically. The upper panel shows a concentration response for a drug with sedative properties. The shape of this response is determined using the E_MAX_ equation. Light sedation is associated with a bispectral index (BIS) of 73. This target effect is associated with a target concentration of 3.7 µg/L. Pharmacokinetic knowledge (lower panel) is then used to achieve this target concentration in the effect compartment (Ce). A 2-compartment pharmacokinetic model is shown in this example. Concentration in the central compartment (Cp) is linked to that in the effect compartment by a rate constant (k1e = keo at steady-state).

**Figure 2 jcm-12-01642-f002:**
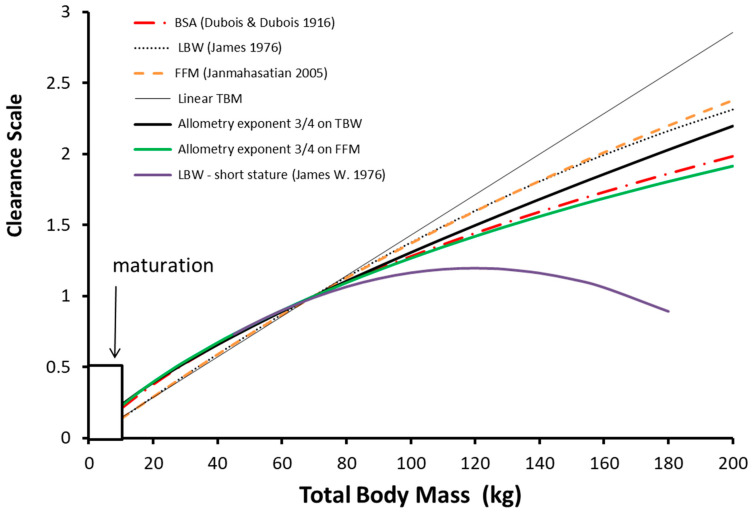
Clearance relative to a person of 70 kg total body mass (TBM) is shown using different size metrics. Children younger than 1 year of age (approx. 10 kg) are not shown because maturation is incomplete in that cohort. Metrics are standardized to a male with typical height for age and weight from 10 to 200 kg. A nonlinear relationship exists between weight and clearance for most body size metrics. The use of the linear per kilogram model, based on TBM, increasingly overestimates clearance in adults of weight greater than 70 kg. The use of BSA (weight with an exponent of 2/3) and allometry using an exponent of 3/4 are similar at lower masses but diverge when TBM is greater than 100 kg. Note that the James formula [57] (purple line) fails in adults of short stature with increasing total body weight [58,59].

**Figure 3 jcm-12-01642-f003:**
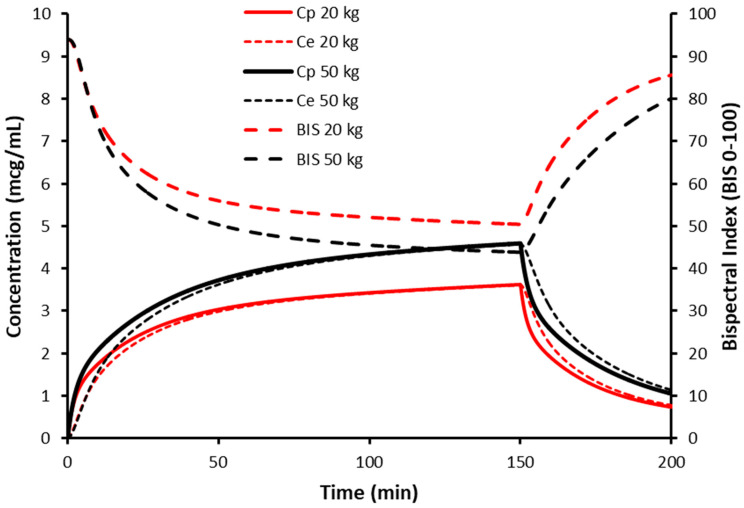
Propofol time concentration and concentration effect profiles after infusion (10 mg/kg/h) in a 6-year-old male child of 20 kg, height 115 cm and a 6-year-old child of 50 kg, height 115 cm. Effect is measured by bispectral index (BIS). Concentrations are higher in the 50 kg child because clearance is less than that in the 20 kg child.

**Figure 4 jcm-12-01642-f004:**
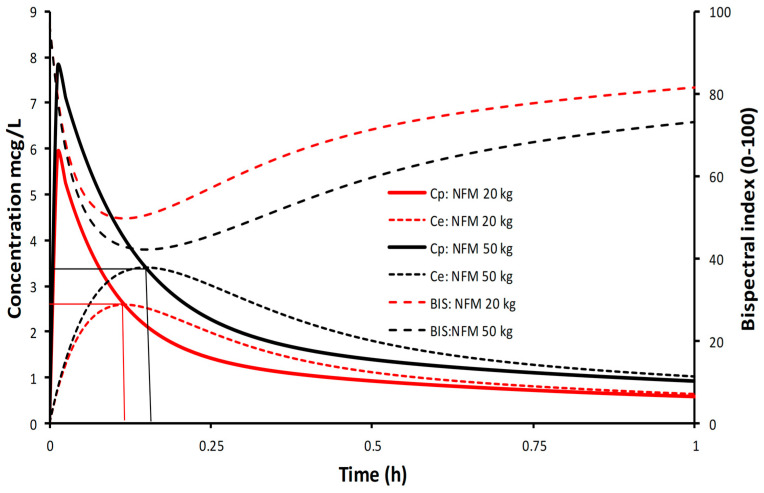
Estimation of volume of distribution at the time of peak effect (Vpe) in a 6 year old child of 20 kg and of 50 kg given dexmedetomidine 1 µg/kg [109]. Simulation performed using dexmedetomidine pharmacokinetic parameters derived by Morse and colleagues [87]. Dexmedetomidine clearance was best scaled using FFM and NFM (FfatV = 0.29) for volume of distribution.

**Figure 5 jcm-12-01642-f005:**
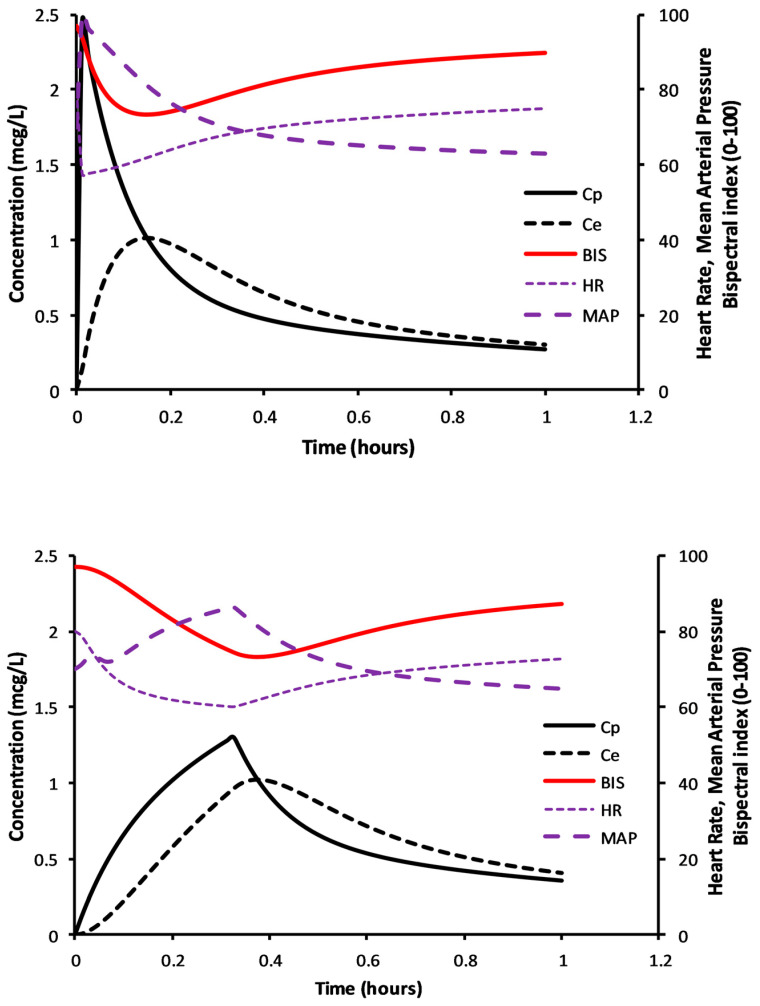
Simulation of hemodynamic adverse effects when dexamethasone is given intravenously to a 6-year-old child (20 kg). Drug in the upper panel was administered as a rapid bolus 0. 9 µg/kg. The lower panel shows hemodynamic changes after a larger loading dose of 1.1 µg/kg delivered as an infusion over 20 min. A similar level of sedation (BIS 73) is achieved at 20 min but the slow infusion is without the dramatic changes in heart rate and mean arterial blood pressure observed after rapid infusion.

**Table 1 jcm-12-01642-t001:** Factor of fat (Ffat) estimates for pharmacokinetic parameters of clearance and volume of distribution for glomerular filtration rate (GFR) and selected drugs that have been investigated.

	Ffat Clearance	Ffat Volume	Source
GFR	0.22	-	Neonates to adults, *n* = 928 [86]
Acetaminophen	0.816	1	Adults 18–49 y, 49–116 kg, *n* = 116 [87]
Busulfan	0.509	0.203	0.1–66 years, *n* = 1610 [88]
Dexmedetomidine	0	0.293	Neonates to adults, *n* = 202 [87]
Dexmedetomidine	0 **	0	Adults Obese *n* = 20, age 18–54 y, Weight 94–152 kg, BMI 36–52 kg·m^−2^ Lean *n* = 20, age 18–60 y, Weight 59–97 kg, BMI 23–30 kg·m^−2^) [89]
Ethanol	1 (Vmax)	0.39	Adults, *n* = 108 [90]
Gemcitabine	0	0	Adults, *n* = 56 [91]
Heparin	0	1	Children 0.5–15 y *n* = 64 [92]
Ibuprofen	0.863	0.718	Adults 18–49 y, 49–116 kg, *n* = 116 [87]
Lithium	0	0	Children (*n* = 61) [93]
Miltefosine	0	0	Children and adults [94]
Oxycodone	1	1	Neonates to adults, *n* = 237 [95]
Oxypurinol	0	0	Adult patients with gout (*n* = 92), healthy subjects (*n* = 12) [96]
Propofol	1	1	Adults obese (*n* = 19, age 40 SD 8.7 y, Weight 106 SD 18 kg, BMI 39.7 SD 4.1 kg·m^−2^) and 51 non-obese (*n* = 51) [46]
Remifentanil (LBM from [57])	0	0	Adults 18–60 years, *n* = 24 Obese 38 SD 8 y, Weight 113 SD 17 kg Lean 38 SD 7 y, Weight 64 SD 10 kg [63]
Tacrolimus	0	0	Adult kidney transplant recipients, *n* = 44 [97]
Warfarin	0	0	Adults, *n* = 264 [98]

** = Fat mass was associated with reduced clearance, suggesting organ dysfunction associated with obesity in adults.

**Table 2 jcm-12-01642-t002:** Dexmedetomidine infusion rate required to achieve a target effect of bispectral index of 73 in a male child of 6 years of age and height of 115 cm. The target concentration was 1 µg/mL. The loading dose was administered over 20 min. Subsequent infusion rates were altered at 20 min and 50 min in order to maintain the target concentration. Simulation for rate determination performed using pharmacokinetic parameters from Morse et al. [87].

	20 kg	30 kg	40 kg	50 kg	20 kg	30 kg	40 kg	50 kg
Duration (min)	Rate: mg/kg/h				Rate mg/h			
0–20	2.9	2.58	2.27	2	58.78	77.4	90.9	100
20–50	1.4	1.14	1.00	0.90	28.0	34.2	40.0	45
50–150	1.1	0.89	0.78	0.69	22.4	26.9	31.0	34.4

**Table 3 jcm-12-01642-t003:** Allometric scaling for Fat-Free Mass and Total Body Mass.

	Mass Scaled Using Allometry	
Total Body Weight (kg)	Total Body Mass (kg)	Fat-Free Mass (kg)
10	16.3	15.8
20	27.4	28.0
30	37.1	37.8
40	46.0	47.0
50	54.4	55.6
60	62.4	62.8
70	70.0	70.0
80	77.4	76.6
90	84.5	82.8
100	91.5	88.7
110	98.3	94.2
120	104.9	99.5
140	117.7	109.3
160	130.1	118.2
180	142.1	126.5
200	153.8	134.1

## Data Availability

No new data were created or analyzed in this study. Data sharing is not applicable to this article.

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
