# Peer review of "Considerations for Intravenous Anesthesia Dose in Obese Children: Understanding PKPD"

_jcm, 2023, doi:10.3390/jcm12041642_

Round 1

Reviewer 1 Report

I would like to thank the authors for the effort to perform this detailed and for sure time consuming review. It reminds the reader how complicated the pharmacodynamic and pharmacocinetic processes are in fact. The review demonstrates very well, that one model does not fit for all drugs. The paper also makes clear, that the effect is not only explained by pharmacological parameters, clearance and volume it is aiso influenced by the cumulative effect of different drugs used for induction and maintenance of anesthesia.

But unfortunately in the end this review does not realy give practical advice how to manage anesthesia in the ebese child.

I am missing a serous discussion of the different models mentioned and at least try to give the reader some practical advice.

Author Response

I would like to thank the authors for the effort to perform this detailed and for sure time-consuming review. It reminds the reader how complicated the pharmacodynamic and pharmacokinetic processes are in fact. The review demonstrates very well, that one model does not fit for all drugs. The paper also makes clear, that the effect is not only explained by pharmacological parameters, clearance and volume it is also influenced by the cumulative effect of different drugs used for induction and maintenance of anesthesia.

But unfortunately in the end this review does not really give practical advice how to manage anesthesia in the obese child. I am missing a serious discussion of the different models mentioned and at least try to give the reader some practical advice.

RESPONSE: We have added a section with practical recommendations. The numerous other size models used in pediatric anaesthesia have been previously reviewed. That review is referenced in the manuscript.1  We have also added Mencken’s famous line to the start of the paper, emphasizing that there is no easy answer to this conundrum!

  1. Anderson BJ, Holford NH. What is the best size predictor for dose in the obese child? Pediatr Anesth 2017;27:1176-84.

Reviewer 2 Report

Thanks for such an exhaustive review of the PKPD models used in TCI systems for TIVA. This was a learning experience for me to review this article. 

As you noted as a limitation of NFM, it is impractical to calculate it in the busy OR. TCI systems are not commonly used in USA secondary to FDA restrictions. We use modified EEG/BIS to help guide the depth of anesthesia and adjust our dosing based on that. What advantage does the complicated math derived calculations provide to the average anesthesiologist?

7.3.4: considerations for adverse events- Hypotension and bradycardia are well known side effects of rapid bolus of dexmedetomidine regardless of the patient's weight hence it is always recommended that the loading dose is given as a slow infusion over 10 min. How is it any different for obese children and for TCI?

Several heading and subheadings have formatting issues that needs to be corrected. (Ex- line 149 major heading is 5. followed by subheading 4.1, then subheading 5.2 etc).

Author Response

Response to Reviewer 2

Thanks for such an exhaustive review of the PKPD models used in TCI systems for TIVA. This was a learning experience for me to review this article. 

As you noted as a limitation of NFM, it is impractical to calculate it in the busy OR. TCI systems are not commonly used in USA secondary to FDA restrictions. We use modified EEG/BIS to help guide the depth of anesthesia and adjust our dosing based on that. What advantage does the complicated math derived calculations provide to the average anesthesiologist?

**Getting the dose right is one of the 5 drug safety mantras (right drug, right dose, right time etc). We now offer some practical advice. We agree that BIS/EEG can be useful, but remains a poor guide to “depth of anaesthesia”. There are some key aspects to this manuscript

  1. To teach readers that this is a complex problem and that there are multiple considerations to “correct dose”
  2. There is no easy solution such as ideal body weight
  3. TCI pumps incorporate PKPD and do the calculations for us practitioners, but practitioners need to know about those PKPD considerations
  4. Practical advice for those without TCI pumps and
  5. The need for monitoring depth of anaesthesia (clinically or pEEG); the “art of anaesthesia” does not go away. That old question from days of past has hot gone away. What is the dose of thiopentone (pentothal)? Answer. Enough. It has just swapper out thiopentone for other drugs

7.3.4: considerations for adverse events- Hypotension and bradycardia are well known side effects of rapid bolus of dexmedetomidine regardless of the patient's weight hence it is always recommended that the loading dose is given as a slow infusion over 10 min. How is it any different for obese children and for TCI?

**The practical advice that dexmedetomidine be given over 10 min or so in adults was determined from practical observations.1 We demonstrate, using simulation, how PKPD can be used to explain these observations and predict what will happen if given administered more slowly (or quickly). It is this sort of information that will be used in TCI pumps for children in the near future. Loading dose will be administered not over 10 min, as in adults, but rather 15-20 min to minimize adverse effects. These adverse effects are concentration related. Dose required in the obese child to achieve that concentration will differ from the lean child. Another impact of dose in the obese is effect on CSHT; we have previously explored that issue.2,3

Several heading and subheadings have formatting issues that needs to be corrected. (Ex- line 149 major heading is 5. followed by subheading 4.1, then subheading 5.2 etc).

**These have been corrected

  1. Dexdor (dexmedetomidine): an overview of Dexdor and why it is authorised in the EU. European Medicines Agency, 2020. 2021, at https://www.ema.europa.eu/en/medicines/human/EPAR/dexdor.)
  2. Morse JD, Cortinez LI, Anderson BJ. Pharmacokinetic concepts for dexmedetomidine target-controlled infusion pumps in children. Pediatr Anesth 2021;31:924-31.
  3. Morse JD, Cortinez LI, Meneely S, Anderson BJ. Propofol context-sensitive decrement times in children. Paediatr Anaesth 2022;32:396-403.

Round 2

Reviewer 1 Report

Thank you for the revision of the paper. This makes the document easier to read. I personally find table 3 more confusing than helpful in clinical work. Furthermore I would suggest, that a native speaker is checking the English spelling once more.

Author Response

Many thanks for these comments.

We have made slight changes to both text and headings of table 3 to improve clarity. We are simply suggesting a fictitious weight for the infusion pump. The idea has been used many times in the past e.g. fictitious height input or even "pharmacokinetic mass"

Shibutani K, Inchiosa MA, Jr., Sawada K, Bairamian M. Accuracy of pharmacokinetic models for predicting plasma fentanyl concentrations in lean and obese surgical patients: derivation of dosing weight ("pharmacokinetic mass"). Anesthesiology 2004;101:603-13.

Two of the authors are native English speakers, but unfortunately practice the "New Zealand" dialect that confuses many. We have used spell check for USA English and found a few minor errors that have been corrected.